# Immunization with Bovine Serum Albumin (BSA) in Oil-Adjuvant Elicits IgM Antibody Response in Chinese Soft-Shelled Turtle (*Pelodiscus Sinensis*)

**DOI:** 10.3390/vaccines8020257

**Published:** 2020-05-29

**Authors:** Cheng Xu, Jiehao Xu, Yu Chen, Øystein Evensen, Hetron Mweemba Munang’andu, Guoying Qian

**Affiliations:** 1Zhejiang Provincial Top Key Discipline of Biological Engineering, Zhejiang Wanli University, Ningbo 315100, China; cheng.xu@nmbu.no (C.X.); xujiehao@zwu.edu.cn (J.X.); yuchen_20@hotmail.com (Y.C.); 2Department of Paraclinical Sciences, Faculty of Veterinary Medicine, Norwegian University of Life Sciences, P.O. Box 369, 0102 Oslo, Norway; oystein.evensen@nmbu.no

**Keywords:** IgM, ELISA, immunoassay, soft-shelled turtle, immunoglobulin

## Abstract

Immunoassays are among the frontline methods used for disease diagnosis and surveillance. Despite this, there are no immunoassays developed for the Chinese soft-shelled turtle (*Pelodiscus sinensis*), which has expanded into large scale commercial production in several Asian countries. One of the critical factors delaying the development of immunoassays is the lack of characterized soft-shelled turtle immunoglobulins. Herein, we used mass spectrometry together with the ProtQuest software to identify the soft-shelled turtle IgM heavy chain in serum, which again was used to produce a polyclonal anti-turtle-IgM in rabbits. Thereafter, the polyclonal anti-turtle-IgM was used as a secondary antibody in an indirect ELISA to evaluate antibody responses of soft-shelled turtles injected with the bovine serum albumin (BSA) model antigen. Our findings show that only turtle immunized with a water-in-oil BSA plus ISA 763A VG adjuvant (SEPPIC, France) emulsion had antibodies detected at 42 days post vaccination (dpv) while turtles injected with phosphate buffered saline (PBS) only as well as turtle injected with BSA dissolved in PBS had no significant antibody levels detected in serum throughout the study period. In summary, our findings show that rabbit polyclonal anti-turtle-IgM produced can be used in ELISA to measure serum antibody responses in immunized soft-shelled turtles. Future studies should explore its application in other immunoassays needed for the disease diagnosis and vaccine development for soft-shelled turtles.

## 1. Introduction

The Chinese soft-shelled turtle (*Pelodiscus sinensis*), formerly known as *Trionyx sinensis*, is widely cultured in China, Japan and Vietnam and more recently it has expanded to Korea, Malaysia, Singapore, Thailand, and Philippines [1]. Outside Asia it has been introduced in Brazil, Hawaii and Spain [2]. Its production has increased exponentially from about 5000 tons in 1994 to more than 300,000 tons in 2016 because of its high reproductive potential, pharmaceutical value, wide consumer acceptance and high demand as food for human consumption [3]. Expansion has been hampered from increasing disease problems. Bacterial species identified to cause diseases in soft-shelled turtles include *Aeromonas hydrophila*, *A. sobria*, *A. veronii*, *Citrobacter freundii*, *Morganella morganii*, *Edwardsiella tarda* and *Pseudomonas aeruginosa* while pathogenic fungi include *Paecilomyces lilacinus*, *Aphanomyces sinensis* and *Saprolegnia* spp. [4,5,6,7,8,9]. Viruses identified to cause diseases in soft shelled turtles include the Soft-shelled turtle iridovirus (STIV) [10], Trionyx sinensis hemorrhagic syndrome virus (TSHSV) [11] and turtle systemic septicemia spherical virus (TSSSV) [12]. To elucidate the prevalence of these pathogens, there is need to develop rapid diagnostic tools for mass surveillance.

The pace at which diagnostic tools are being developed is slower than the rate at which novel pathogens infecting the soft-shelled turtle are being discovered. There is a general lack of immunoassays required for the diagnosis of pathogens causing diseases in the soft-shelled turtle. Immunoassays are analytical assays that rely on binding between an antigen and antibody, and there are three Ig isotypes found in turtles namely IgM, IgD and IgY [13]. In the Chinese soft-shelled turtle, the cDNA sequence of the Ig joining (J) chain has been cloned and characterized in which the deduced amino acid sequence has a high homology with previously reported turtle J chain (80.7%) and of chicken (71.3%). Significant upregulation of the J-chain transcripts quantified by RT-PCR were observed in spleen, kidney and blood of turtles vaccinated against *A. hydrophila* pointing to the importance of the J chain to immunization [14]. In another study, Xu et al. detected IgM, IgD and IgY mRNA in the Chinese soft-shelled turtle using qRT-PCR of which the detected IgM and IgY constant domains are similar with other vertebrate species [15]. Yet, these studies only identified the different Ig and J chain mRNA transcripts, but the proteins involved have not been characterized. The lack of characterized immunoglobulins from soft-shelled turtle has likely delayed the development of immunoassay for diagnosis of pathogens infecting the Chinese soft-shelled turtle and assessment of immune responses post infection or immunization. 

The objective of the present study was to isolate and characterize the Chinese soft-shelled turtle IgM heavy chain using mass spectrophotometry followed by immunization of rabbits to raise a polyclonal anti-turtle IgM antibody. In addition, we aimed to use the developed polyclonal anti-turtle IgM antibody in an ELISA to test antibody responses in soft-shelled turtle immunized with the bovine serum albumin (BSA) as the model antigen. Ultimately, a polyclonal anti-turtle IgM antibody is then used in immunoassays for disease diagnosis and for use in evaluating antibody responses to vaccination. 

## 2. Material and Methods

### 2.1. Animals

Healthy Chinese soft-shelled turtle (*Pelodiscus sinensis*) weighing 60 g on average were purchased from a Chinese soft-shelled turtle farm in Zhejiang Province, China in June 2019 and were put to acclimatize for 2 weeks at 25 °C in freshwater in tanks before starting the experiments. They were fed daily with commercial diet (Foshan Haihuang, China) and maintained in freshwater at 25 °C in tanks, where water was changed weekly during the acclimatization and experimental period. No attempt was made to separate the sexes. All animal experiments were carried out following the guidelines approved by the Institutional Animal Care and Use Committee at Zhejiang Wanli University in China (Ethic approval code 0520001).

### 2.2. Isolation of Soft-Shelled Turtle IgM 

Ten Chinese soft-shelled turtles were anesthetized with tricaine methane-sulfonate (MS-222) (Sigma-Aldrich) and euthanatized. Blood was collected, and serum was separated from blood by centrifuging at 2500 rpm for 15 min. Serum was pooled from 10 turtles for IgM isolation giving a total volume of approximately 10 ml. Briefly, the same volume of saturated solution of ammonium sulphate (NH_2_SO_4_) (4.1 M at 25 ℃) was added to the pooled serum for antibody precipitation. The precipitate was resuspended in phosphate buffered saline (PBS) and dialyzed in PBS for 36 h, and PBS was changed every 6 h. The sample after dialysis was centrifuged at 4500 rpm for 20 min, and the supernatant was subjected to a second round of NH_2_SO_4_ precipitation and dialysis process before the isolated antibodies were analyzed by SDS-PAGE.

### 2.3. SDS-PAGE and Mass Spectrometry Analysis

After IgM isolation, the proteins were separated by 12% SDS-PAGE gel under reduction conditions of which the electrophoretic run was performed at 200 V for 90 min. Coomassie brilliant blue R250 (Sigma-Aldrich, St. Louis, MO, USA) was used to visualize the protein bands in the polyacrylamide gel electrophoresis (PAGE). Thereafter, the protein bands above 70 kDa were excised from the polyacrylamide gel and subjected to in-gel tryptic digestion followed by liquid chromatography electrospray ionization mass spectrometry (LC-ESI-MS) analysis. Briefly, the digested protein samples were analyzed by a high-pressure liquid chromatography (HPLC) system (Agilent, Palo Alto, CA, USA), followed by mass spectrometry MS using a linear ion trap mass spectrometer (LTQ, Thermo, San Diego, CA, USA). The mass spectrometric data were used to search against the UniProt protein database with ProtTech’s ProtQuest (Philadelphia, PA, USA) software suite.

### 2.4. Production of Rabbit Anti-Turtle IgM Polyclonal Antibodies

After identification of the protein bands by mass spectrometry, the band containing the IgM protein was excised and sent to HuaAn Biotechnology Co., Ltd. (Hangzhou, China) for immunization of rabbits to produce the polyclonal anti-turtle IgM antibody. Briefly, one healthy New Zealand white rabbit, weighing 2.5 kg, was primed with 0.5 mg IgM protein formulated in FCA and boosted 3 times with 0.25 mg IgM protein formulated in FIA at 2-week intervals. Serum was collected 2 weeks after final immunization, and the rabbit polyclonal anti-IgM antibodies were characterized by Western blot (WB) analysis.

### 2.5. Model Antigen, Vaccine Formulation and Immunization

The bovine serum albumin (BSA) used as a model antigen was purchased from Sangon Biotech Co., Ltd. (Shanghai, China) and dissolved in PBS at the concentration of 0.5 mg/mL and kept frozen at −20 °C until use. The water-in-oil (W/O) emulsions containing the BSA model antigen was prepared by mixing 70% adjuvant Montanide ISA 763A VG (Seppic, Paris, France) (w/w) with 30% aqueous antigen (w/w) according to manufacturer’s guidelines (Seppic, Paris, France). The final BSA concentration in the (W/O) formulation was 0.5 mg/mL. Thereafter, 30 Chinese soft-shelled turtles were injected intraperitoneally with 0.2 mL (100 µg/turtle) of BSA dissolved in PBS (BSA/PBS-Group), and kept in 3 separate tanks maintained at 25 °C. Another 30 Chinese soft-shelled turtles were injected intraperitoneally with 0.2 mL (100 µg BSA/turtle) of the BSA water-in-oil emulsion (w/o) (BSA/Adjuvant-Group) and kept in 3 separate tanks maintained at 25 °C. Another 24 Chinese soft-shelled turtles were injected intraperitoneally (i/p) with 0.2 mL PBS (PBS-Group) and served as non-vaccinated controls. Eight individuals were sampled at day 0 (prior to vaccination) and served as reference. At 14, 28, and 42 days post-vaccination (dpv), 8–10 soft-shelled turtles from the BSA/PBS and BSA/adjuvant groups were used for blood collection (Figure 2) while from the PBS-group 8 turtles were collected at 14, 28, and 42 dpv for blood collection. All turtles used for sample collection were anesthetized using MS-222 and euthanatized, blood was drawn, and serum was separated by centrifuging at 2500 rpm for 15 min and stored at −20 °C until antibody analysis by ELISA.

### 2.6. Chinese Soft-Shelled Turtle Antibody Analysis by Indirect ELISA

For turtle antibody analysis by ELISA, 96-flat bottom micro-titer plates (Nunc Maxisorb, Roskilde, Denmark) were coated with 100 µL 20 µg/mL BSA diluted in coating buffer (0.05 M carbonate–bicarbonate buffer, pH 9.6) and incubated at 4 °C overnight. Thereafter, the plates were washed three times with TBST (0.02 M Tris–HCl, 0.9% NaCl, 0.05% Tween 20, pH 7.6) and blocked with 300 µL 5% dry milk in TBST for 2 h at room temperature. This was followed by adding 100 µL sera diluted 1:50 with 1% dry milk in TBST in duplicates to the wells and incubation for 1 h. After washing the plates five times using TBST, 100 µL rabbit anti-turtle IgM diluted 1:400 was added to the wells and incubated for 1 h. Thereafter, plates were washed five times followed by adding horseradish peroxidase (HRP) conjugated goat anti-rabbit Ig (Sangon Biotech Co., Ltd., Shanghai, China) diluted 1:2000 to each well and the plates were incubated for 1 h. After the final wash, substrate tetramethylbenzidine (TMB) was added to each well. After color development within 15 min, the reaction was stopped by adding 50 µL of 1 M H_2_SO_4_ and the optical density (OD) was measured at 450 nm wavelength using a Microplate spectrophotometer reader (Hangzhou Allsheng Instruments, Hangzhou, China).

### 2.7. Statistical Analysis

All statistical analyses of ELISA results were performed using GraphPad Prism version 5.00 for Windows (GraphPad Software, San Diego, CA, USA). Comparisons of antibody responses between different immunization groups was done using a Kruskal Wallis test and Stata15 for analysis. Differences were considered statistically significant at *p* < 0.05 (Confidence limits 95%).

## 3. Results 

### 3.1. Characterization of Soft-Shelled Turtle IgM 

Figure 1 (lane A) shows SDS-PAGE analysis of the isolated proteins including the IgM heavy chain from soft-shelled turtle serum. The expected 70 kDa plus protein band containing the IgM heavy chain was characterized by mass spectrometry analysis. The different proteins identified using the ProtTech’s ProtQuest software by blasting in the UniProt protein database is shown in Table 1, where molecular weight (MW), number of unique peptides identified for each protein and relative abundance of each protein identified are shown. The major constituents detected include serotransferrin (TF), inter-alpha-trypsin inhibitor heavy chain H3 (ITIH3), alpha-2-macroglobulin (α2M)-like protein, immunoglobulin M (IgM) heavy chain constant region, serum albumin (sAlb) and transferrin receptor protein 1 (TFRC). Note that the largest proportion of the protein band detected was TF (71.6%) followed by the IgM heavy chain constant region (7.8%), ITIH3 (4.1%), sAlb (3.4%), α2M (1.8%) and TFRC (1.3%) accounting for approximately 90.0% of the total isolated proteins above 70 kDa detected from the soft-shelled turtle serum. The isolated turtle IgM heavy chain including constant and variable region, with expected molecular weight of 70 kDa plus is shown by western blot (WB) using the anti-turtle IgM raised in rabbit. Note that while WB analysis showed that the IgM heavy chain with a constant and variable region had a molecular weight of 70 kDa plus (Figure 1, lane B), Table 1 shows that the IgM heavy chain constant region without variable region had a molecular weight estimated to be 56.9 kDa. This is mainly because the IgM reference sequence detected on UniProt did not have the variable region sequences, but only had the heavy chain constant region and hence the molecular weight detected by mass spectrometry had a lower MW of 56.9 kDa that excludes the variable region.

### 3.2. Detection of Antibodies to BSA Model Antigen by ELISA

Figure 2 shows a summary of ELISA data of antibody levels from turtles immunized with the BSA model antigen in which the rabbit polyclonal anti-turtle IgM produced in this study was used as a secondary antibody. The mean OD-levels detected before vaccination at 0 dpv from the unvaccinated turtles was OD_450_ = 0.238 (standard error = 0.012, *n* = 8), being not significantly different from OD-levels detected at 14 dpv in the PBS (OD_450_ = 0.24, SE = 0.010, *n* = 8), BSA-PBS (OD_450_ = 0.246, SE = 0.008, *n* = 8) and BSA-adjuvanted (OD_450_ = 0.231, SE = 0.009, *n* = 8) groups indicating that there were no specific antibodies detected in response to immunization in any of the groups at 14 dpv. Similarly, there was no significant increase in OD-levels detected in the PBS (OD_450_ = 0.252, SE = 0.008, *n* = 6), BSA-PBS (OD_450_ = 0.336, SE = 0.046, *n* = 8, p = 0.37) and BSA-adjuvanted (OD_450_ = 0.377, SE = 0.10, *n* = 9, p = 0.21) groups at 28 dpv. In contrast, antibody levels detected in the BSA-adjuvant group at 42 dpv (OD_450_ = 0.580, SE = 0.12, *n* = 9) were significantly higher (p = 0.029) than the PBS group (OD_450_ = 0.197, SE = 0.007, *n* = 8) and significantly higher (p = 0.0268) than the BSA-PBS group (OD_450_ = 0.247, SE = 0.017, *n* = 8). 

## 4. Discussion

In this study, we have identified the soft-shelled turtle IgM heavy chain protein by mass spectrometry and excised it to produce a rabbit polyclonal anti-IgM antibody for use in ELISA. To the best of our knowledge, this is the first report on production of the rabbit anti-soft-shelled turtle IgM and its use to measure antibody responses by ELISA. A previous study by Xu et al. [15] detected the soft-shelled turtle IgM mRNA in blood, kidney and spleen immunized with an inactivated *A. hydrophila* vaccine using qRT-PCR. The major limitation with qRT-PCR detection methods is that they do not quantify the specific protein expressed by the host, and will not always reflect the magnitude of the host response to antigen stimulation. Further, antibodies are widely used as a gold standard to measure host response to vaccination in efficacy trials and they are also used as a diagnostic measure of post exposure to infection [16,17]. 

Adjuvants are immune potentiators known to increase the level and duration of antibody responses. In the present study, the BSA-adjuvanted group had high IgM levels at 42 dpv and significantly lower in the non-adjuvanted BSA-PBS group. Time of immune induction used in the present study was chosen from a previous study [15] that had peak IgM mRNA levels at 35 dpv followed by a sharp decline to insignificantly low levels at 42 dpv in soft-shelled turtles immunized with an inactivated *A. hydrophila* vaccine. Here we found by ELISA that serum antibody levels in the adjuvanted BSA group were highest at 42 dpv suggesting that antibody levels were still on an increase at this time. The kinetics of humoral responses show that a latent phase occurs between the time when B- and T-cells first contact a novel antigen to proliferation and differentiation to produce antibodies. In general, this latent phase is shorter in birds being three to five days [18] while in mammals it lasts about one to two weeks [19]. Zimmerman et al. [20] showed that in reptiles the latent phase lasts about six to eight weeks when IgM levels begin to increase to reach peak levels being in line with our observations that IgM levels were higher at 42 dpv than at earlier sampling time points. Another important factor shown to influence the kinetics of antibody responses in reptiles is seasonal changes in immune function mainly influenced by temperature changes [21]. As pointed out by Zimmerman [21], the immune response of ectotherm is often depressed after prolonged exposure to cold. Grey [22] showed that painted turtles (*Chrysemus pincta*) injected with 25 mg BSA reared at 22 to 25 °C had antibodies detected at six weeks post vaccination in the group injected with BSA plus adjuvant, which is in line with our findings that the soft-shelled turtle cultured at 25 °C in the present study developed antibodies in the BSA-adjuvant group being the same temperature range (22 to 25 °C) and duration of six weeks post vaccination for antibodies increase in both Grey’s [22] and the present study. Nonetheless, future studies should seek to determine the peak and duration of antibody responses using post vaccination samples collected over a longer period after vaccination than what was used in the present study. Differences between adjuvanted and non-adjuvanted vaccine preparations should also be studied in more detail. 

The use of mass spectrometry for identification and characterization of antibodies has been described by several scientists [23,24], and as pointed out by Ladwig et al. [24], when using mass spectrometry to identify and characterize antibodies, the heavy chain variable region is the target and its uniqueness is compared with existing antibody repertoires. Similarly, in the present study we used the ProtTech’s ProtQuest software followed by blasting in the UniProt protein database to show that the 70 kDa plus protein excised from the polyacrylamide gel contained the IgM heavy chain peptides that match with the existing soft-shelled turtle IgM heavy chain constant region sequences. The neutralizing ability of soft-shelled turtle antibodies against bacteria has previously been tested using the serum agglutination test by Cheng-zhu et al. [25], and they found that serum from soft-shelled turtles immunized with *A. hydrophila* lipopolysaccharide (LPS) agglutinated live bacteria and blocked its replication in vitro [25]. Suffice to point out that various serum constituents contribute to the agglutination of pathogens and these include complement, TF, α2M, sAlb and various Ig isotypes [26,27,28] being similar with several proteins detected by mass spectrometry from the soft-shelled turtle serum in the present study. Therefore, it is likely that these proteins could have contributed to the agglutination of bacteria reported in the previous studies [25]. Here we focused on determining the antibody levels of serum IgM after immunization by ELISA excluding several other serum agglutinants. In summary, our findings show that the rabbit polyclonal anti-turtle-IgM made in this study can reliably be used to measure antibody responses induced in soft-shelled turtles. Future studies should seek to optimize the ELISA developed herein and to explore the application of the rabbit polyclonal anti-turtle-IgM in other immunoassays for the diagnosis of various infectious diseases and vaccine development for the Chinese soft-shelled turtle. 

## 5. Conclusions

In this study we have used mass spectrometry to identity the soft-shelled turtle IgM heavy chain that we used to produce a polyclonal anti-turtle IgM in rabbit. Using the polyclonal anti-turtle IgM produced herein as a secondary antibody in an indirect ELISA, we have shown that soft-shelled turtle injected with BSA-adjuvant produced antibodies that increased by 42 dpv being in line with a previous study done by Grey [22] who detected antibodies in painted turtles (*Chrysemus pincta*) injected with BSA-adjuvant reared at 22 to 25 °C after 6 weeks post immunization. Put together, these findings suggest that turtles exposed to the same temperature range respond to antigen stimulation in a similar pattern resulting in antibody production at same rate. Overall, we have shown that the rabbit polyclonal anti-turtle antibody produced herein can reliably be used in ELISA to measure antibodies produced in response to immunization in soft-shelled turtle. We advocate that future studies should validate the ELISA developed herein for use in diagnosis of infectious diseases and vaccine development.

## Figures and Tables

**Figure 1 vaccines-08-00257-f001:**
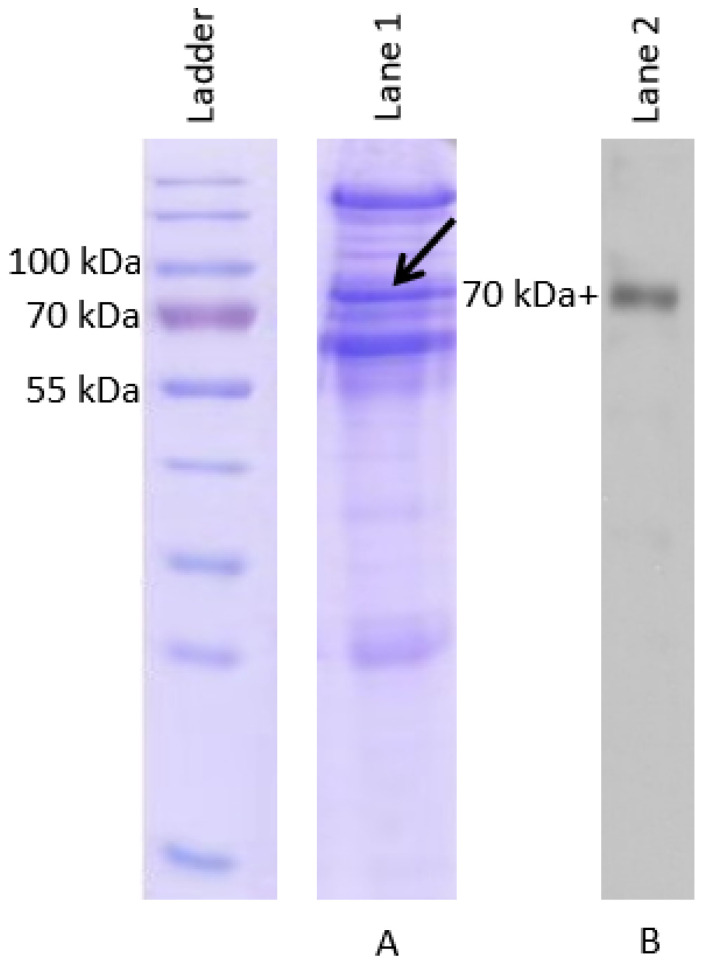
SDS-PAGE analysis of isolated Chinese soft-shelled turtle immunoglobulin (**A**) followed by Western blot (**B**) using anti-turtle IgM antibody raised in rabbit. The expected molecular weight of soft-shelled turtle IgM heavy chain is 70 kDa plus as indicated by the arrow in the figure.

**Figure 2 vaccines-08-00257-f002:**
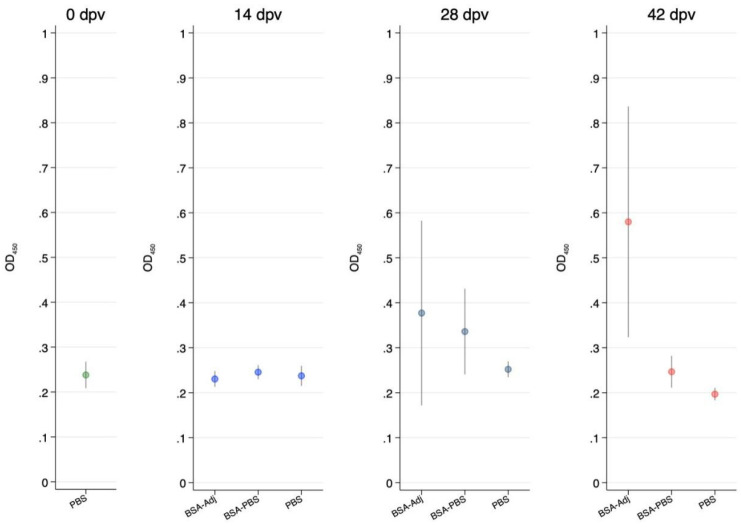
Serum anti-IgM antibody response of Chinese soft-shelled turtles at 0, 14, 28 and 42 days post vaccination (dpv) for phosphate buffered saline (PBS), bovine serum albumin (BSA)/PBS and BSA/adjuvant groups. Average and 95% confidence limits are shown for each group at each time post vaccination.

**Table 1 vaccines-08-00257-t001:** Mass spectrometry results of protein bands ≥70 kDa excised from the gel electrophoresis.

Hits	Protein Name	Abbr	GeneBank Acc. no.	Molecular Weight (kDa)	Number of Unique Peptides	Relative Abundance	Probability
1	Serotransferrin	TF	XP_014429958	81.5	164	71.6%	99.0%
2	Inter-alpha-trypsin inhibitor heavy chain H3	ITIH3	XP_006127649	100.7	62	4.1%	99.0%
3	Alpha-2-macroglobulin-like	α2M	XP_014425635	151.8	36	1.8%	99.0%
4	Immunoglobulin M heavy chain constant region	IgM	ACU45376	56.9	35	7.8%	99.0%
5	Serum albumin	sAlb	XP_006125464	71.0	28	3.4%	99.0%
6	Transferrin receptor protein 1	TFRC	XP_006112414	84.3	20	1.3%	99.0%
7	Complement C5	C5	XP_006120838	189.1	11	0.2%	99.0%
8	Alpha-fetoprotein	AFP	XP_006125463	69.5	9	0.8%	99.0%
9	Complement C3	C3	XP_006124830	186.3	5	0.1%	99.0%
10	Kininogen-2-like	KNG2	XP_006138593	87.3	4	0.3%	99.0%

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
