# Peer review of "Immunization with Bovine Serum Albumin (BSA) in Oil-Adjuvant Elicits IgM Antibody Response in Chinese Soft-Shelled Turtle (Pelodiscus Sinensis)"

_vaccines, 2020, doi:10.3390/vaccines8020257_

Round 1

Reviewer 1 Report

The manuscript "Immunization with bovine serum albumin (BSA) in oil-adjuvant elicits IgM antibody response in Chinese soft-shelled turtle (Pelodiscus sinensis)" describes the first approach on assessing the antibody response in an unconventional farmed species such as Chinese soft- shelled turtle.

The authors used bovine serum albumin as an antigen, adjuvanting it with ISA 763A VG in a water in oil emulsion.

The experimental procedure adopted by the authors, as well as the description made in the text, does not highlight any criticality; in fact, the experimental design appears congruous and the description of both the preliminary immunization steps and the antibody assessment phases are sufficiently described and well argued.

The results are clearly inserted and the discussion is well reasoned and clear.
The only note that I would like to make in this paper is the lack of a reference on the authorization to experiment and on the response of the ethics committee for the actions that have been taken.

Apart from this note that absolutely must be included in the text in full and well detailed, in my opinion, the work can be published after minor revision.

Author Response

QUERY-1: The only note that I would like to make in this paper is the lack of a reference on the authorization to experiment and on the response of the ethics committee for the actions that have been taken. Apart from this note that absolutely must be included in the text in full and well detailed, in my opinion, the work can be published after minor revision.

RESPONSE: We have inserted a response that all experiments were carried with the approved guidelines of the Institutional Animal Care and Use Committee at Zhejiang Wanli University in China (see lines 71-78)

Reviewer 2 Report

Immunization with bovine serum albumin (BSA) in oil-adjuvant elicits IgM antibody response in Chinese soft-shelled turtle (Pelodiscus sinensis)

Cheng Xu et al.

This is a well constructed and executed study and a well-crafted paper. The authors have identified the soft-shelled turtle IgM heavy chain in serum and produced a polyclonal anti-turtle-IgM from rabbits. Turtles were challenged with BSA and the anti-turtle-IgM was used to evaluate antibodies with ELISA. Antibodies were detected at 42 days.

Methods section.

Since reptiles can have seasonal variability in immune function, can the authors provide information about the time of year that these studies were performed? Could this affect reproducibility in other laboratories?

There are some issues that can be addressed in the discussion.

For example, the authors found that the IgM heavy chain protein had a molecular weight of 70 kDa, but mass spectrometry identified the IgM heavy chain constant region with molecular weight of 56.9 kDa. Can the authors speculate on the reason for the difference?

Some readers may not understand the difference between reptile immune response versus mammals. The authors could include a statement or two about the temporal length of IgM production in reptiles, for example, H.M. Grey; Phylogeny of the Immune Response;Studies on Some Physical Chemical and Serologic Characteristics of Antibody Produced in the Turtle; J Immunol, 1963, 91 (6) 819-825; Zimmerman LM, Vogel LA, Bowden RM. Understanding the vertebrate immune system: insights from the reptilian perspective. J Exp Biol. 2010;213(5):661‐671. doi:10.1242/jeb.038315. This would help in understanding the detection of IgM at 42 days.

Author Response

QUERY-1: Since reptiles can have seasonal variability in immune function, can the authors provide information about the time of year that these studies were performed? Could this affect reproducibility in other laboratories?

RESPONSE: The seasonality variability effect on immune function has now been explained in lines 209 and 222 in the current submission. Thanks to the reviewer for pointing out this aspect which had overlooked. We find our data in agreement with references provided (see lines 209 to 222).

QUERY-2: For example, the authors found that the IgM heavy chain protein had a molecular weight of 70 kDa, but mass spectrometry identified the IgM heavy chain constant region with molecular weight of 56.9 kDa. Can the authors speculate on the reason for the difference?

RESPONSE: We explained that Mass spectrometry data shown in Table-1 does not include the variableregion while the western blot result shows the IgM heavy chain constant region plus variable region. An explanation is made in lines 161 to 167.

QUERY-3: Some readers may not understand the difference between reptile immune response versus mammals. The authors could include a statement or two about the temporal length of IgM production in reptiles, for example, H.M. Grey; Phylogeny of the Immune Response; Studies on Some Physical Chemical and Serologic Characteristics of Antibody Produced in the Turtle; J Immunol, 1963, 91 (6) 819-825; Zimmerman LM, Vogel LA, Bowden RM. Understanding the vertebrate immune system: insights from the reptilian perspective. J Exp Biol. 2010;213(5):661‐671. doi:10.1242/jeb.038315. This would help in understanding the detection of IgM at 42 days.

RESPONSE: We have provided more detail to explain that reptiles have a longer induction phase than birds and mammals when B- and T-cells begin to proliferate and differentiate after first encounter with a novel antigen lasting up-to 6 – 8 weeks for IgM levels to increase. References have been provided to support this observation (see lines 209 to 222).